# Improving the Enzymatic Activity and Stability of a Lytic Polysaccharide Monooxygenase

**DOI:** 10.3390/ijms24108963

**Published:** 2023-05-18

**Authors:** Miesho Hadush Berhe, Xiangfei Song, Lishan Yao

**Affiliations:** 1Qingdao New Energy Shandong Laboratory, Qingdao Institute of Bioenergy and Bioprocess Technology, Chinese Academy of Sciences, Qingdao 266101, China; miesho@qibebt.ac.cn (M.H.B.);; 2Shandong Energy Institute, Qingdao 266101, China; 3University of Chinese Academy of Sciences, Beijing 100049, China; 4Department of Biotechnology, College of Natural and Computational Sciences, Aksum University, Axum 1010, Ethiopia

**Keywords:** lytic polysaccharide monooxygenases (LPMOs), protein engineering, cellulose, depolymerization, enzyme activity and stability, synergism

## Abstract

Lytic Polysaccharide Monooxygenases (LPMOs) are copper-dependent enzymes that play a pivotal role in the enzymatic conversion of the most recalcitrant polysaccharides, such as cellulose and chitin. Hence, protein engineering is highly required to enhance their catalytic efficiencies. To this effect, we optimized the protein sequence encoding for an LPMO from *Bacillus amyloliquefaciens* (*Ba*LPMO10A) using the sequence consensus method. Enzyme activity was determined using the chromogenic substrate 2,6-Dimethoxyphenol (2,6-DMP). Compared with the wild type (WT), the variants exhibit up to a 93.7% increase in activity against 2,6-DMP. We also showed that *Ba*LPMO10A can hydrolyze *p*-nitrophenyl-β-D-cellobioside (PNPC), carboxymethylcellulose (CMC), and phosphoric acid-swollen cellulose (PASC). In addition to this, we investigated the degradation potential of *Ba*LPMO10A against various substrates such as PASC, filter paper (FP), and Avicel, in synergy with the commercial cellulase, and it showed up to 2.7-, 2.0- and 1.9-fold increases in production with the substrates PASC, FP, and Avicel, respectively, compared to cellulase alone. Moreover, we examined the thermostability of *Ba*LPMO10A. The mutants exhibited enhanced thermostability with an apparent melting temperature increase of up to 7.5 °C compared to the WT. The engineered *Ba*LPMO10A with higher activity and thermal stability provides a better tool for cellulose depolymerization.

## 1. Introduction

Cellulose, consisting of β-1, 4-linked subunits of glucose molecules, is the most abundant natural resource on the earth, and plays a pivotal role in biorefinery [1]. Cellulose is the major component of the polymetric matrix of lignocellulosic plant biomass, which must be deconstructed into easily degradable smaller units by cellulases prior to biological utilization. Traditional cellulase enzyme cocktails mainly consist of glucoside hydrolases (GH), which include exoglucanases, endoglucanases, and β-glucosidase [2], that breakdown the glycoside bond mainly via hydrolysis. However, the catalytic capabilities of these traditional glycoside hydrolase (GH) enzymes in the crystalline region of cellulose are low, thus, inhibiting the depolymerization efficiency on cellulose [3]. This is mainly due to the recalcitrant nature of the cellulose molecule. Hence, enhancing enzyme catalytic efficiency and reducing the cost of hydrolytic enzymes is a major challenge in the industrial application of enzymes [4]. As recently discovered oxido-reductase copper dependent enzymes, lytic polysaccharide monooxygenases (LPMOs) have promising features for further improvement of the depolymerization process [5]. These enzymes have the ability to boost the activity of cellulose-degrading hydrolytic enzymes. In addition to this, LPMOs contribute significantly to the deconstruction of highly crystalline cellulose via a synergistic mode of action with the commercially available cellulases [6].

Groundbreaking research on LPMOs in 2010 significantly changed our perception of the enzymatic deconstruction of the most recalcitrant compounds, such as cellulose and chitin [7]. Due to their potential to augment the efficiency of hydrolytic enzymes in the depolymerization of lignocellulosic biomass, LPMOs represent a crucial component in industrial enzymatic applications [8,9]. LPMOs are powerful enzymes that play a pivotal role in the enzymatic deconstruction of cellulose through the oxidative breakdown of the scissile glycosidic bond [10,11]. The active site of LPMOs is defined by a copper ion-bound N-terminal histidine, a characteristic feature of a “histidine-brace”, which is unique in nature and this gives LPMOs the potential to be a powerful oxidative enzyme [12,13]. Copper-dependent LPMOs are oxidoreductase metalloenzymes that initiate the depolymerization of plant biomass, which substantially enhances their synergistic efficiency with other enzyme cocktails such as cellulases to promote hydrolytic efficiency in a wide range of crystalline polysaccharides, including cellulose [14].

LPMOs are classified as auxiliary activity (AA) families. Thus far, based on the sequence collected from the carbohydrate-active enzyme (CAZy) database, they have been categorized into eight auxiliary activity (AA) families [15], namely, AA9, AA10, AA11, AA13, AA14, AA15, AA16, and AA17, on the basis of their sequence similarities. They may be active in various types of polysaccharide such as cellulose, chitin, starch and other oligosaccharides. Hence, LPMOs play a great role in our planet’s carbon cycle [16,17]. Among these, AA9 and AA10 are widely studied families for cellulose deconstruction [15].

The LPMO reaction begins with reduction of the active-site copper from Cu [18] to Cu (I) by a variety of external electron donors, such as ascorbic acid, gallic acid, and lignin [19]. LPMOs oxidize the cellulose molecule either at the C1 or C4 carbon position of the scissile glycosidic bond [20]. However, some less regio-selective LPMOs give mixed C1 and C4-oxidized products. The hydroxylation at the C1-carbon position leads to the formation of 1,5-δ-lactones, which are further hydrated into aldonic acids. Meanwhile, the oxidation of the C4-carbon gives 4-ketoaldoses, which are further hydrated into gemidiol [21]. Like the hydrolytic cellulases, LPMOs also have the potential to form an association with carbohydrate-binding modules (CBMs) via a linker [22]. In the beginning, it was thought that the LPMO catalysis reaction strictly depends on molecular O_2_ and the reducing agents. However, recent studies have revealed that H_2_O_2_ can also drive the LMPO catalytic reaction, which is faster than the O_2_-dependent reaction [10,23].

In the last decade, LPMOs have been one of the core research areas in biomass processing and have garnered significant scientific interest for industrial applications. This is mainly due to their ability to promote the depolymerization process of the most recalcitrant crystalline cellulose, and their potential for boosting oxidative cleavage of the crystal structure of cellulose in synergy with commercial cellulases [7,16]. Despite having been widely studied since their discovery and the crucial role that they have in industrial bio refining, so far, very little work has been documented on the hallmarks of industrial enzymes, such as improved activity and stability at a wide temperature range [17,24]. Enhanced enzyme activity and thermostability are the two commonly applied useful traits for industrial enzyme applications [5]. Protein engineering via site mutation is an indispensable aspect of achieving these hallmarks of industrial enzymes, which will subsequently promote the industrial utilization of LPMOs. Site-directed mutagenesis is a powerful approach that has been applied to the optimization of various relevant enzyme traits such as activity improvement, thermal resistance, solvent stability, and regioselectivity [25,26].

In this study, we attempted to engineer a cellulose-active *Ba*LPMO10A from the bacteria *Bacillus amyloliquefaciens*. The main aim of this study is to enhance the enzymatic activity and thermal stability of purified enzymes towards various lignocellulosic substrates. LPMO enzymes are attracting significant attention due to their role in the enzymatic conversion of biomass. Therefore, the enhancement of activity and stability through various available protein engineering tools will be an indispensable approach to their various industrial applications. Generally, the results of the current study showed promising features of both enhanced enzyme activity and better thermal stability for *Ba*LPMO10A mutants, which will subsequently increase the potential of its industrial applications.

## 2. Results

### 2.1. Selection of BaLPMO10A Mutants through Sequence Consensus

Protein sequences of *Ba*LPMO10A and its homologs with an identity between 50.0% and 66.1% were identified through a Blast search. A mutational site was selected if the amino acid type in the wild-type *Ba*LPMO10A was different from the most probable amino acid among all the sequences (Appendix A). The variants were generated by substituting the targeted amino acids with the highly conserved amino acid residues via sequence consensus. A total of 15 single mutants (including Y3F, I4V, K5E, N36A, N36E, V40A, V40E, V40I, V40L, V40M, R55Q, N60G, H77R, E124D, and G158A) were designed for further enzyme activity analysis.

### 2.2. Expression and Screening of BaLPMO10A Mutants

The gene sequences, which encode the WT BaLPMO10A and single mutants, were successfully cloned, expressed in E. coli, and purified (Appendix A). The 15 designed single variants were tested against the soluble substrate 2,6-DMP. The 2,6-DMP-based activity detection is a straightforward, sensitive, and robust method [27,28,29]. Among the 15 tested single mutants, 10 of them showed enhanced activity towards the soluble substrate 2,6-DMP compared to the wild type (WT) (Figure 1A). Particularly, the single mutants V40L, I4V, Y3F, and E124D showed 90%, 82%, 81%, and 80% activity enhancement, respectively (Figure 1A). The structural analysis of BaLPMO10A revealed that many of the high-activity mutational sites, such as Y3, I4, N36, and V40, are near the active site, but some of the high-activity mutational sites, such as K5, R55, and E124, are away from the active site (Appendix A). Furthermore, 20 double mutants were designed by combining single mutants with good activity. All the double mutants designed were examined against the soluble substrate 2,6-DMP, and 11 mutants exhibited higher activity than the WT (Figure 1B). Accordingly, the selected mutants showed up to a 93.7% increase (from V40L-E124D) in activity compared with the WT (Figure 1B). It appears that double mutants have only slightly better activity than individual single mutants. No synergistic effect was seen from different single mutants. Thus, no triple or quadruple mutants were prepared. The apparent kinetic parameters of the purified enzymes were determined using the 2,6-DMP substrate. To this effect, most of the variants showed high values of *k*_cat_ in comparison with the wild type. Hence, higher *k*_cat_ values were observed in the variants I4V-E124D, V40L-E124D, and V40L, with an up to 2-fold increased *k*_cat_ value compared to the wild type (Figure 1C). On the other hand, some of the variants had higher *K_m_* values than the wild type. However, fluctuating and lower *K_m_* values were also observed in some variants (Figure 1D). This might be due to the introduction of some structural changes around the substrate binding site of the enzyme, which perturbs the substrate binding affinity of the enzyme. The single and double mutants that showed higher activity towards the soluble substrate 2,6-DMP were selected for further activity measurement with insoluble substrates such as PASC, Avicel, and FP.

### 2.3. Hydrolysis Activity of BaLPMO10A

Previously, it has been shown that *Ba*LPMO10A has C1 oxidation activity against chitin but no activity against cellulose [30]. Earlier studies have shown that some LPMOs can hydrolyze cellulose or carboxymethylcellulose (CMC) [18,31], albeit with low activity. To test whether *Ba*LPMO10A has hydrolysis activity, CMC was used as a substrate and mixed with WT *Ba*LPMO10A (Figure 2A). To ensure that no oxidation reduction reaction occurred, the reducing reagent ascorbic acid was not added. After 24 h of reaction, ~0.1 mM of reduced sugar was released. Hydrolysis activity was further confirmed using *p*-nitrophenyl-β-D-cellobioside (PNPC) as the substrate (Figure 2B). The hydrolysis of PNPC released 4-nitrophenol, which has distinctive absorption at 405 nm. Both experiments indicate that *Ba*LPMO10A is capable of hydrolyzing the C_1_−O_4_ glycosidic bond.

### 2.4. Action of the Purified BaLPMO10A Enzymes on PASC

The cellulose depolymerization potential of the purified LPMOs alongside the high-activity mutants was assessed against the insoluble substrate Phosphoric Acid Swollen Cellulose (PASC). The amount of reduced sugars produced was evaluated using a detection assay that employs para-hydroxybenzoic acid hydrazide (PAHBAH) [32]. Using PASC as the substrate, it is shown that after 12 h of reaction, the single mutants produced reduced sugar in quantities from 0.19 mM to 0.26 mM, about 19 to 60% higher than the WT (Figure 3A). The single mutants V40L, E124D, and I4V showed the highest increase in reduced sugar production. The higher activity of the single mutants is consistent with the activity assay against the soluble compound 2,6-DMP. For the double mutants, similar activity enhancement was seen. The highest reduced sugar amounts were produced from the double mutants V40L-E124D, I4V-E124D, and V40L-I4V, corresponding to 1-, 0.9-, and 0.8-fold increases, respectively, in the production of reduced sugar compared with the WT.

The amount of reduced sugar released in PASC was compared to the specific activity against 2,6-DMP (Figure 3B). It is known that cellulose glycosidic bond cleavage is quite complex. The enzyme needs to bind to the cellulose surface, search for the glycosidic bond, cleave the glycosidic bond, and release from the surface. It is interesting to see the linear correlation between the activity against 2,6-DMP and PASC, although two different kinds of reaction were performed using *Ba*LPMO10A. The enzyme activity against PASC is so low that even after 12 h of reaction, the product concentration is still very small (less than 0.4 mM, Figure 3A). The inhibition and reverse reaction are probably negligible. The enzyme has a rather high Tm (see discussion below) and appeared to be stable during the reaction. These factors may have an impact on the correlation.

### 2.5. The Synergistic Depolymerization Potential of the Purified Enzymes with Cellulase against Various Substrates

To see whether mutants with high activity assist with cellulose hydrolysis by cellulase, a commercial cellulase cocktail from *Trichoderma reesei* was used to hydrolyze PASC, FP, and Avicel together with *Ba*LPMO10A and different mutants. As shown in Figure 4A,B in the depolymerization of PASC, the combination of the WT *Ba*LPMO10A and the commercial cellulase from *T. reesei* increased the production of reduced sugars 0.8-fold after 12 h of reaction. In comparison, the addition of the mutants to the cellulase increased the production rate up to 1.9-fold. The highly active single mutants V40L, E124D, and I4V increased the release of reduced sugars 1.9-, 1.8-, and 1.7-fold, respectively, compared with the cellulase alone. Similarly, the double mutant V40L-E124D increased the production of the reduced sugars 2.7-fold, followed by the mutants I4V-E124D, V40L-I4V, and V40L-Y3F, which contributed 2.5-, 2.4-, and 2.3-fold increases, respectively, compared with cellulase alone.

In addition to PASC, we also examined the glycosidic bond cleavage of FP by purified *Ba*LPMO10A and the high-activity mutants with synergistical action using the commercial cellulase from *T. reesei*. Compared to PASC, FP has more crystalline cellulose content, and thus, is more difficult to depolymerize. After 48 h of reaction, most single mutants, except V40I, produced ~90 to 140% more reduced sugar than the cellulase alone (Figure 4C). Furthermore, the WT increased the release of the reduced sugars by 30% compared to the production of the reduced sugar by cellulase alone. The four most active single mutants in 2,6-DMP oxidation, V40L, Y3F, I4V, and E124D, also showed higher activity against FP than the others (Figure 4C). In comparison, the double mutants appeared to be slightly more active. The double mutants manifested an up to 2-fold increase in the production of reduced sugar compared with the commercial cellulase from *T. reesei* after 48 h of reaction. The double mutant V40L-E124D increased the rate of production of the reduced sugars 2-fold, followed by V40L-I4V and I4V-E124D, which each enhanced the release of reduced sugars ~1.9-fold. The other three double mutants, V40L-Y3F, V40L-N36E, and E124D-Y3F, also showed higher efficiency in the production of reduced sugars, corresponding to a 1.8-fold increase in each of the mutants compared to that of the commercial cellulase alone (Figure 4D).

Meanwhile, we also evaluated the synergistic action of purified *Ba*LPMO10A together with the commercial cellulase in the degradation of Avicel cellulose (Figure 4E). The WT *Ba*LPMO10A combined with the commercial cellulase from *T. reesei* increased the production of reduced sugars 0.6-fold compared with the production by cellulase alone. The single mutants V40L, I4V, and E124D increased the release of reduced sugar 1.4-, 1.2-, and 1.0-fold, respectively. The same effect can be seen for the double mutant. V40L-E124D significantly increased the production rate 1.9-fold, while the other double mutants, I4V-E124D and E124D-Y3F, increased the yield of the reduced sugar 1.8- and 1.7-fold, respectively, compared to the production of reduced sugars by the commercial cellulase alone.

### 2.6. Enzyme Thermostability from CD Measurements

The thermostability of the WT and the variants was also investigated by measuring the folding–unfolding equilibrium using Circular Dichroism (CD) spectra (Figure 5). The WT begins to unfold in the early stage with a melting temperature of 57.3 °C, whereas the variants begin to unfold at higher temperatures and exhibit enhanced thermostability with the melting temperature increased by up to 7.5 °C compared to the WT (Table 1). Generally, the introduction of mutations improves not only the activity but also the thermostability of LPMOs. The factors that affect enzyme stability and activity are quite complex and need further investigation to fully unravel the underlying mechanisms [33]. It is important to identify the optimum temperature of LPMOs to improve the saccharification process of the cellulosic substrate [34]. The current study suggests that *Ba*LPMO10A mutations improve enzyme thermostability without affecting the optimal temperature. Previously, it was reported that the engineering of LPMOs via site-directed mutagenesis from *Aspergilus fumigatus* showed high stability and activity from 50 to 60 °C [35].

## 3. Discussion

The sequence consensus method is a powerful method of generating enzyme mutants with higher activity. As demonstrated in this work, most of *Ba*LPMO10A single mutants selected using the consensus method show enhanced oxidation activity against 2,6-DMP (Figure 1). Previously, it has been shown that *Ba*LPMO10A is capable of oxidizing chitin but not cellulose [30]. However, hydrolysis activity was detected in this work when using CMC and PNPC as the substrate, although the activity appears to be low (Figure 2). The increase in reduced sugar concentration when using PASC as the substrate further confirms the enzyme hydrolysis activity (Figure 3A). It is also interesting that the hydrolysis activity against PASC is correlated with the oxidation activity against 2,6-DMP. We speculate that the hydrolysis and oxidation activities are performed by the same copper binding active site.

Although the hydrolysis activity of *Ba*LPMO10A (and its mutants) against PASC is rather low, it shows a synergistic effect with the commercial cellulase from *T. reesei* when they are mixed together (Figure 4) [36]. A similar synergistic effect can be seen when using FP and Avicel as the substrate. FP and Avicel have larger amounts of crystalline regions than PASC, and thus, are more difficult to hydrolyze. The synergistic effect is also quite obvious, although the hydrolysis activity of *Ba*LPMO10A is expected to be lower for these two substrates. Furthermore, the *Ba*LPMO10A mutants with higher activity tend to be more effective at cellulose degradation when combined with the commercial cellulase, implying that hydrolysis by *Ba*LPMO10A is complementary to that by the commercial cellulase. It has been indicated that the LPMO reaction is crucial to enhancing the degradation of microcrystalline cellulosic substrates. A study reported the enhanced depolymerization of a pre-treated corn stover where the addition of LPMOs to the cellulase mixture led to a reduction in half of the enzyme load required to reach 90% of the hydrolysis process of cellulose [31]. Moreover, another more recent report showed that the addition of LPMOs to the cellulase resulted in an up to 50% increase in the saccharification process efficiency of the microcrystalline cellulose [37]. In another study, in most of the LPMO-catalyzed reactions, the combined effect with commercial cellulase was mainly derived from the oxidative cleavage of lignocellulosic substrates [38]. However, in this work, the result is different, in that the synergistic effect arises from the hydrolytic activity of *Ba*LPMO10A. It is not surprising that *Ba*LPMO10A mutants with a higher activity can have a better synergistic effect with the commercial cellulose than the WT.

The thermostability of *Ba*LPMO10A and its mutants was studied using a CD spectrometer (Figure 4). The WT *Ba*LPMO10A has a Tm of 57 °C (Table 1). Most single and double mutants show better thermostability, with the highest Tm reaching about 64 °C. In the enzyme sequence consensus analysis, the replacement of the amino acid by the most frequent one is usually beneficial for the enzyme whether the mutation increases enzyme stability or activity needs to be verified experimentally. In the case of *Ba*LPMO10A, it appears that both thermostability and enzyme oxidation and hydrolysis activity are improved through sequence consensus analysis.

The commercial usage of LPMOs involves the utilization of harsh and unnatural conditions, necessitating the need for enzyme optimization, despite the fact that LPMOs have an extraordinary capacity to function on highly crystalline substrates. To depolymerize polysaccharides, enzyme cocktails are often added after a high-temperature pre-treatment is used to process the lignocellulosic biomass. It is widely acknowledged that the thermostability of the enzymes in these cocktails is a characteristic that has economic significance; this is because stable enzymes allow for higher reaction temperatures, which lower cooling costs, improve reactant solubility, and lower the risk of microbial contamination [39].

There is now a vast array of naturally occurring and engineered thermostable cellulase; however, only few works have been reported on the improvement of the thermostability of LPMOs through protein engineering so far [40]. Recently, work reported by Zhou et al. showed promising results in the enhancement of the enzymatic activity and stability of LPMOs through the introduction of a disulfide bond [33]. Our work provides an example of LPMO engineering through the sequence consensus method. The improved enzyme provides a better tool for cellulose degradation.

## 4. Materials and Methods

### 4.1. Materials and Chemical Reagents

Unless stated otherwise, all the materials and chemical reagents used in this study were procured from Yuanye Bio-Technology Co., Ltd. (Shanghai, China), Sinopharm Chemical Reagent Co., Ltd. (Beijing, China), Sigma-Aldrich chemicals Pvt. Ltd. (St. Louis, MO, USA), and Thermo Fisher Scientific (Waltham, MA, USA).

### 4.2. Cloning, Expression, and Purification of BaLPMO10A

The DNA sequence that encodes for the WT and variants of *Ba*LPMO10A, PDB code (2YOX), which has a 6*His tag at its C-terminus and a SUMO tag at its N-terminus, was ligated with the vector pET-20b containing the restriction enzymes NdeI and XhoI. Then, the construct was transformed into *E. coli* BL21 (DE3) (Novagen, Merck, Darmstadt, Germany) to enable the expression of the proteins, as described elsewhere [41,42]. All the variants were generated via a PCR-based site-directed mutation with the primers listed (Appendix A), and were verified via DNA sequencing. The WT and all the variants were expressed and purified in a similar way. Briefly, a fresh colony was inoculated into a 3L Luria–Bertani (LB) medium containing 100 µg/mL Ampicillin [43]. The cells were grown at 37 °C in a vigorously shaking incubator (~220 rpm) until the optical density of the culture at 600 nm reached 0.6–0.8. Then, a 1 mM concentration of isopropyl-β-D-thiogalactopyranoside (IPTG) was added to induce the expression of recombinant proteins at 18 °C for 24 h. The cells were harvested via suspension in 50 mM sodium acetate (pH = 5.0), lysed via ultrasound sonication, and centrifuged (9600 g, 4 °C, 30 min.). Then, the resulting supernatant was filtered using a syringe filter with 0.45 µm pores. After this, the proteins were purified in a pre-equilibrated Ni-NTA column using the ÄKTA pure chromatography system (GE Healthcare, Chicago, IL, USA). The non-target proteins bound on the column were washed with washing buffer (50 mM sodium acetate, 20 mM Na_2_HPO_4_, 300 mM NaCl, and 10 mM Imidazole). Finally, the target proteins were eluted by applying a linear gradient of imidazole (20–500 mM). The purity of the proteins was checked by subjecting them to sodium dodecyl sulfate polyacrylamide gel electrophoresis (SDS-PAGE). The purified proteins were concentrated using an Amicon ultracentrifuge filter with a molecular weight cutoff of 10 KDa (Merck Millipore, Schwalbach, Germany). After this, the purified proteins were dialyzed to change the buffer to 50 mM sodium acetate with a pH of 6.0 prior to storage at −20 °C. To obtain the native N-terminus, the purified proteins (WT and all the variants) were digested with SUMO protease (Novagen, Billerica, MA, USA) as described previously [44]. After cleavage, the SUMO tag was removed using Xarrest agarose beads (Novagen). Moreover, the fusion proteins were loaded onto a Ni-NTA affinity chromatograph (Novagen) for further purification. The concentration of the purified proteins was determined by measuring their optical density at 280 nm using UV−vis spectroscopy (Nanodrop 2000, Thermo Fisher) [45]. The coefficient of extinction of the purified proteins was 44,920 M^−1^ cm^−1^ at 280 nm.

### 4.3. Copper Saturation

The purified *Ba*LPMO10A proteins were copper-saturated according to a previously described method [46], with some modifications. Briefly, a purified *Ba*LPMO10A protein (3 mg/mL) was saturated with an excessive amount of copper sulfate (4-fold molar excess) in 20 mM Tris-HCl buffer (pH 8.0) by incubating it at room temperature for about 30 min. After this, the excess copper was removed using a gravity protocol and a PD 10 MidiTrap G-25 (GE Healthcare), a desalting column equilibrated with 50 mM sodium acetate buffer with a pH of 6.0.

### 4.4. Site-Directed Mutagenesis

In order to investigate the effects of various amino acid substitutions at different positions of the parent strain, site-directed mutagenesis was conducted by applying the previously reported method [47]. All the mutants were made via PCR-based site-directed mutagenesis and were verified via DNA sequencing. Then, all the PCR products were expressed in *E. coli* BL21 (DE3).

### 4.5. Enzyme Activity Assay

The enzymatic activity of the purified *Ba*LPMO10A was assayed using the chromogenic molecule 2, 6-dimethoxyphenol (2,6-DMP) (J & K chemicals Co., Ltd. (Shanghai, China) as the main substrate and H_2_O_2_ as the co-substrate, according to the recently proposed spectrophotometric detection method [27]. Briefly, an enzymatic reaction was carried out in 100 mM sodium acetate buffer (pH = 6.0), 100 µM H_2_O_2_, 10 mM 2,6-DMP, and 0.5 µM copper-loaded *Ba*LPMO10A (enzymes) at 30 °C for 5 min. The increase in absorbance at 469 nm was measured against a blank solution that contained the same enzymatic reaction mixture components (except the enzyme) incubated under the same experimental conditions. The coefficient of extinction (ε469 = 53,200 M^−1^ cm^−1^) was taken to determine the concentration of the colored products produced during the reaction time [27].

### 4.6. Hydrolysis of PNPC and CMC by the Purified Enzyme

To examine the hydrolysis potential of the purified enzyme against PNPC, a reaction was carried out in 50 mM sodium acetate buffer at a pH of 6.0, 5 µM *Ba*LPMO10A, 1 mM PNPC. The reaction mixture was incubated at 50 °C over periods of 0, 6, 12, 24, 36, and 48 h. The reaction was terminated by adding 1M Na_2_CO_3_. The amounts of products formed during the reaction were determined by measuring the absorbance at 405 nm using a microplate reader (Ultrospec visible plate reader II 96, GE Healthcare BioScience). Finally, a *p*-nitrophenol standard curve was used to quantify the final concentration of the products released from the reaction [48]. Moreover, the degradation potential of the purified enzyme against CMC was examined. The reaction mixture consisted of 0.5% *w/v* CMC and 5 µM *Ba*LPMO10A, and was incubated in a 1.5 mL Eppendorf tube with a total reaction volume of 250 µL in 50 mM sodium acetate buffer with a pH of 6.0. The reaction was carried out at 50 °C over periods of 0, 6, 12, 24, 36, and 48 h. The reaction was terminated by passing the reaction mixture through a filter with a pore size of 0.22 µm. The final concentration of the products formed was sing buy the PAHBAH assay method [32].

### 4.7. The Depolymerization of PASC by the Purified Enzymes

To evaluate the activity of the purified enzymes in the degradation of PASC, a reaction mixture consisting of 0.5 *w/v* PASC, 1 µM copper-saturated *Ba*LPMO10A (WT and mutants), and 2 mM ascorbic acid was incubated in a 1.5 mL Eppendorf tube with a total reaction volume of 250 µL, in 100 mM sodium acetate buffer with a pH of 6.0. The reaction was carried out at 50 °C over periods of 1, 2, 4, 6, 8, 10, and 12 h. The reaction was terminated by passing the reaction mixture through a filter with a pore size of 0.22 µm. The final concentration of the products formed was determined using the PAHBAH assay method [32].

### 4.8. Synergistic Effect of the Purified Enzymes with the Commercial Cellulases on the Depolymerization of Cellulosic Substrates

To examine the combined effect of the purified enzymes and cellulase from *T. reesei,* we evaluated it against PASC, FP, and Avicel. PASC was generated from Avicel (Microcrystalline cellulose, Sigma-Aldrich PH101, St. Louis, MO, USA) as previously described by [49,50]. Synergism of the purified enzymes (WT and the mutants) and the cellulase from *T. reesei* against the PASC substrate was carried out in 100 mM sodium acetate buffer with a pH of 6.0, 2 mM ascorbic acid, 0.5 W/V PASC, 0.25 mg/mL cellulase (*T. reesei*), and 1 µM copper-saturated *Ba*LPMO10A (WT and mutants) in a total volume of 250 µL in 1.5 mL Eppendorf tube. It was incubated at 50 °C over periods of 1, 2, 4, 6, 8, 10, and 12 h. The product was determined using the PAHBAH assay method [32]. Moreover, the same experimental conditions were used while investigating the synergistic effect of the purified enzymes and cellulase on the depolymerization of Avicel (10 g/L), except the periods of incubation were 6, 12, 24, 36, and 48 h, and the enzyme concentration was 2 µM.

Moreover, to assess the activity of the purified enzymes in the degradation of filter paper, a reaction mixture consisting of 10 g/L filter paper (FP, Whatman No. 1, Ge Healthcare Limited Buckinghamshire, UK), 2 µM copper-saturated *Ba*LPMO10A (WT and mutants), 0.25 mg/mL cellulase (*T. reesei*), and 2 mM ascorbic acid was incubated in a 2 mL Eppendorf tube with a total reaction volume of 400 µL, in 100 mM sodium acetate buffer with a pH of 6.0. The reaction of the mixture was carried out at 50 °C over periods of 6, 12, 24, 36, and 48 h. The products formed during the enzymatic hydrolysis of FP by the purified enzymes were mainly reduced sugars (cellulose) [51].

### 4.9. Thermostability of the Purified Enzymes from CD Measurements

The thermostability of the purified enzymes (WT and the variants) were investigated by measuring the Circular Dichroism (CD) spectra using a Chirascan™ CD Spectrometer equipped with a temperature-regulated cell holder (Applied Photophysics, Leatherhead, Great Britain) as previously described by [52]. The equipment was filled with nitrogen at a flow rate of 10 L min^−1^ during the entire experimental process. The temperature of the reaction was set to 20–95 °C. The reaction of the mixture was carried out in 50 mM sodium acetate buffer with a pH of 6.0. The CD values were plotted using the Origin 8 software and fitted with a four-parameter Sigmoidal curve [52].

## 5. Conclusions

In conclusion, we investigated the protein engineering of an LPMO via site-directed mutagenesis. The activity and stability of LPMO variants were compared with the wild type. The purified enzymes were assayed using the chromogenic soluble substrate 2,6-DMP. Accordingly, some of the mutants showed higher activity than the WT. In addition to this, the potential of the purified enzymes was assessed in the depolymerization of the cellulosic substrates PNPC, CMC, and PASC. Furthermore, the synergistic effect of the purified enzymes (WT and the mutants) was examined in combination with the commercial cellulase from *T. reesei* in the degradation of PASC, FP, and Avicel. The combination of LPMO and cellulase significantly augmented the hydrolysis of the crystalline cellulose compared to cellulase alone. The thermostability of the purified enzymes was evaluated by measuring the CD spectra at different temperatures. The mutants showed higher apparent melting temperature values, and thus, were more stable than the WT. The findings of this study demonstrate that protein engineering through the sequence consensus method is effective at improving LPMOs. Further investigations are required to fully unravel the underlying mechanisms of the improvement of enzyme activity and stability, which will then be further improved to pave the way for their practical utilization in industrial enzyme cocktails.

## 6. Patents

The authors declare the following competing financial interest(s): a Chinese patent has been filed using parts of the results given in this paper.

## Figures and Tables

**Figure 1 ijms-24-08963-f001:**
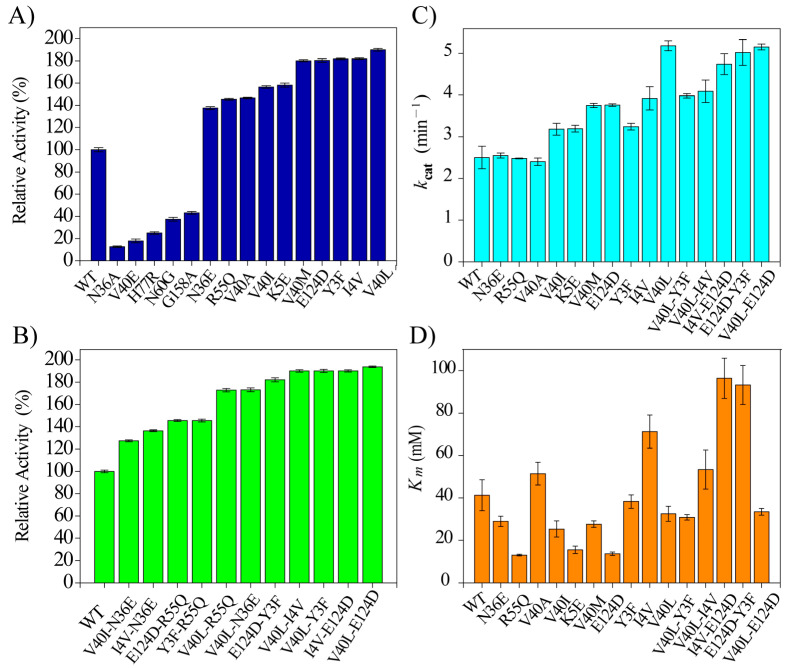
Relative enzyme activity against the soluble substrate 2,6-DMP: (**A**) the WT and single mutants; (**B**) WT and the double mutants. (**C**) Apparent *k_cat_* values. (**D**) Apparent *K_m_* values. The reaction mixture contained 100 mM sodium acetate buffer at pH of 6.0, 0.5 µM *Ba*LPMO10A, 10 mM 2,6-DMP, and 100 µM H_2_O_2_. The optical absorbance was measured at 469 nm for 5 min at 30 °C, and linear regression was used to fit the slope, and thus, obtain the reaction rate (**A**,**B**). *k_cat_* and *K_m_* were derived via fitting of the reaction rate at different 2,6-DMP concentrations (Appendix A). The error bar shows the standard deviation of independent experiments performed in triplicate.

**Figure 2 ijms-24-08963-f002:**
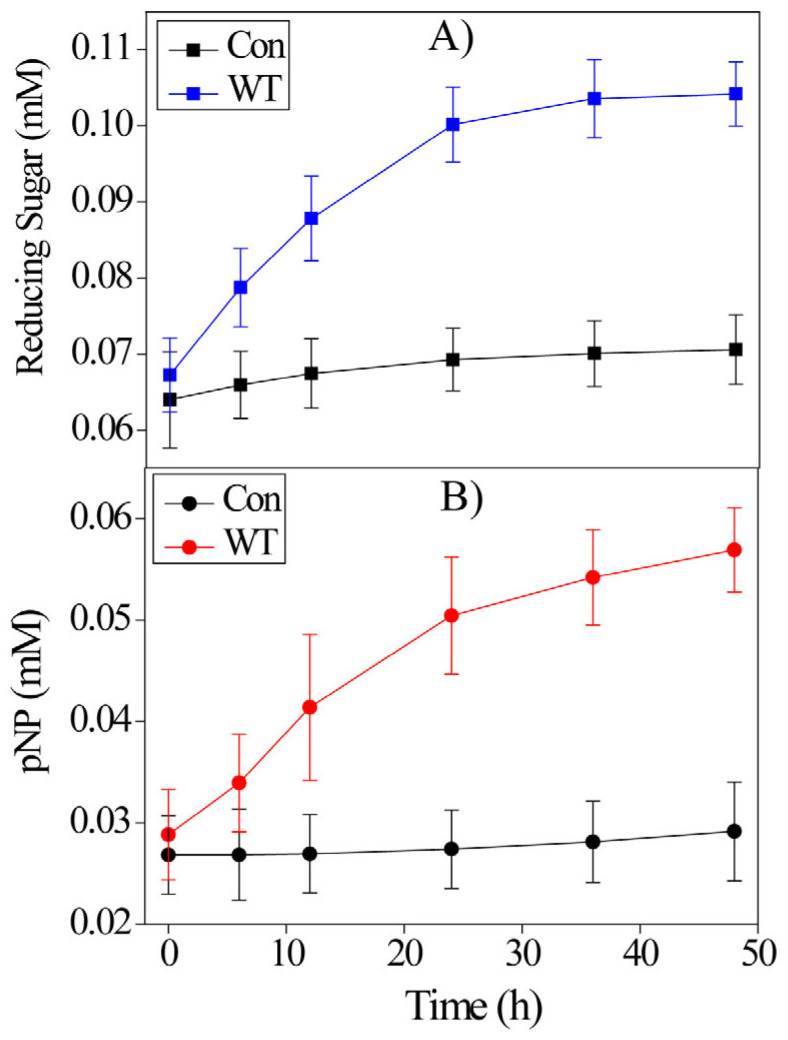
Hydrolysis activity of the purified WT *Ba*LPMO10A against (**A**) CMC (0.5% *w/v*) and (**B**) PNPC (1 mM). The reaction mixture contained 5 µM *Ba*LPMO10A and 50 mM sodium acetate buffer at pH of 6.0. The control (Con) experiment was carried out without adding the enzyme. Error bars represent the standard deviation of three independent experiments.

**Figure 3 ijms-24-08963-f003:**
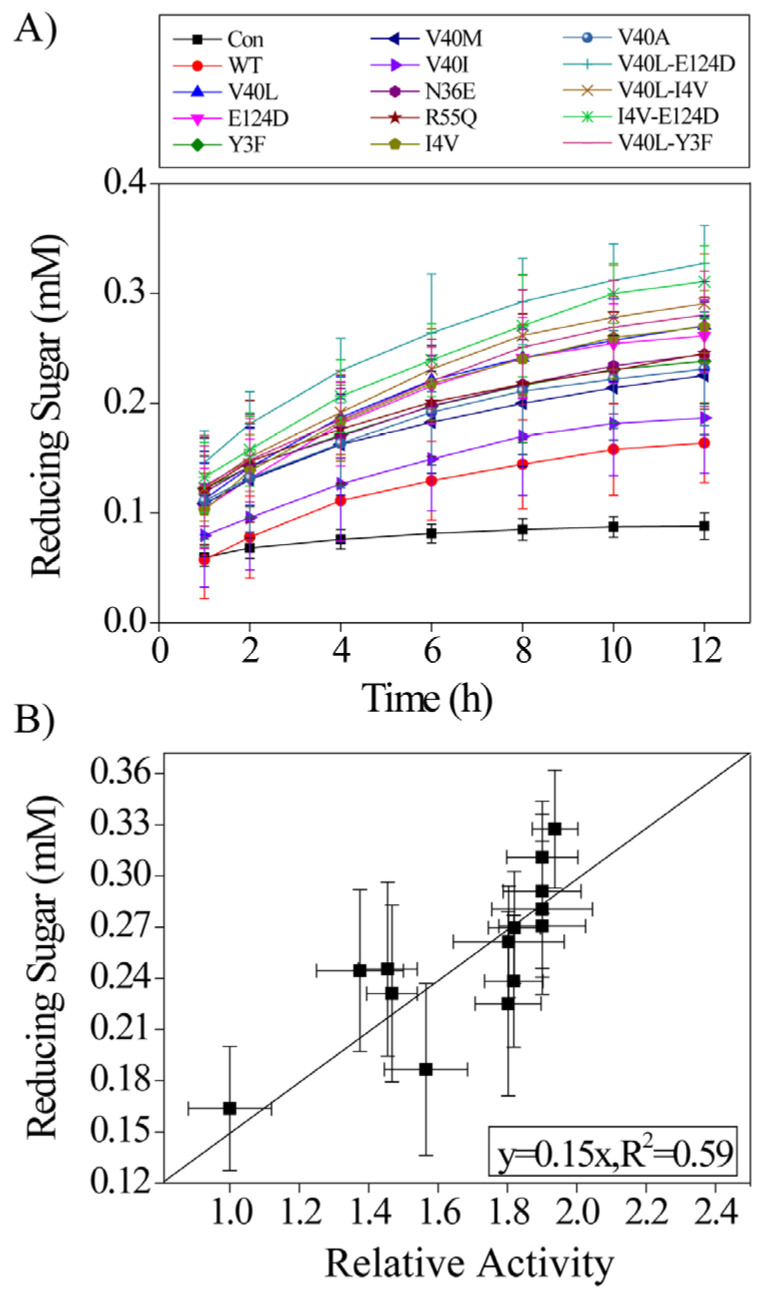
(**A**) Time course of PASC hydrolysis by *Ba*LPMO10A. The reaction mixture contained 100 mM sodium acetate buffer at pH of 6.0, 1 µM *Ba*LPMO10A, 2 mM ascorbic acid, and 0.5% *w/v* PASC. The incubation temperature was 50 °C. The control (Con) experiment was carried out without adding the enzyme. (**B**) Linear correlation between the activity against the soluble substrate 2,6-DMP (the x axis) and that against the insoluble substrate PASC (the y axis). Error bars represent the standard deviation of three independent experiments.

**Figure 4 ijms-24-08963-f004:**
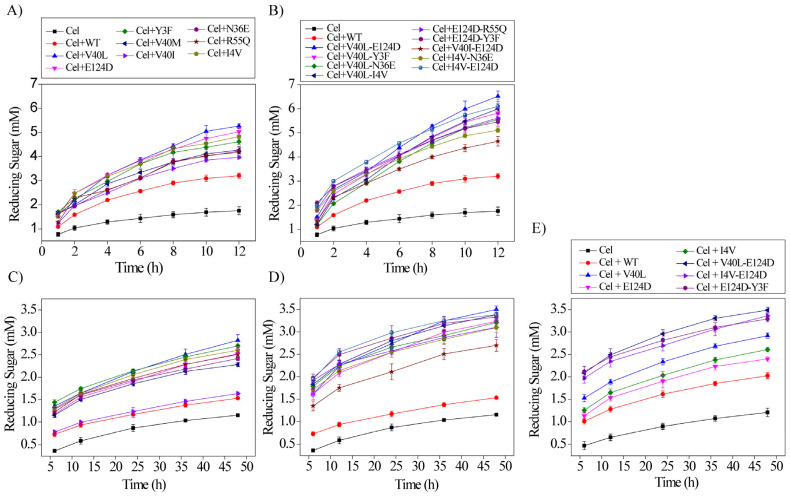
Synergistic action of the purified enzymes with the commercial cellulase (Cel) from *T. reesei*: (**A**) PASC with the single mutants, (**B**) PASC with the double mutants, (**C**) FP with the single mutants, (**D**) FP with the double mutants, (**E**) Avicel with single and double mutants. The reaction mixture contained 100 mM sodium acetate buffer at pH of 6.0, 0.25 g/L cellulase from *T. reesei*, 1 µM *Ba*LPMO10A (for 0.5% *w/v* PASC), 2 µM *Ba*LPMO10A (for 10 g/L FP), 2 µM *Ba*LPMO10A (for 10 g/L Avicel), and 2 mM ascorbic acid. The incubation temperature was 50 °C. Error bars represent the standard deviation of three independent experiments.

**Figure 5 ijms-24-08963-f005:**
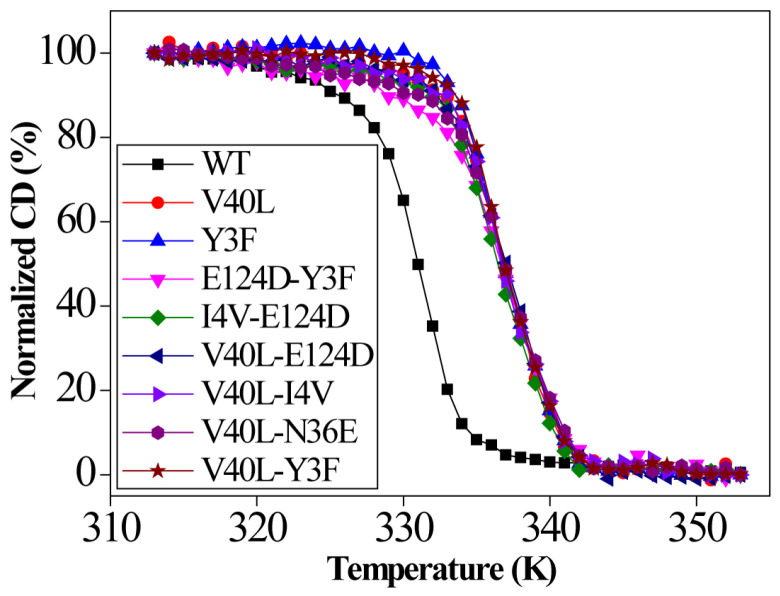
CD spectra of the wild type (WT) and some selected mutants. The experiment was carried out in 50 mM sodium acetate buffer at pH of 6.0; enzyme concentration was 0.1 mg mL^−^^1^.

**Table 1 ijms-24-08963-t001:** Apparent Tm values for the WT and all the variants determined from CD measurements.

*Ba*LPMO10A	*T*_m_ (°C)	*Ba*LPMO10A	*T*_m_ (°C)
WT	57.30 ± 0.12	V40M	63.10 ± 0.12
V40I-Y3F	55.25 ± 0.22	I4V-E124D	63.80 ± 0.20
V40I-E124D	59.30 ± 0.08	Y3F	63.90 ± 0.22
V40I-R55Q	59.90 ± 0.20	V40L-Y3F	64.10 ± 0.19
V40L-R55Q	60.20 ± 0.24	E124D	63.14 ± 0.10
E124D-R55	60.35 ± 0.10	V40L-I4V	63.90 ± 0.25
N36E	60.45 ± 0.15	V40L	64.20 ± 0.20
Y3F-R55Q	60.85± 0.09	V40L-Y3F	64.30 ± 0.28
K5E	62.60 ± 0.37	V40L-N36E	64.57 ± 0.22
V40I	62.80 ± 0.11	V40L-E124D	64.82 ± 0.20
I4V	62.90 ± 0.18		

## Data Availability

The original contributions presented in this study are included in the article. Further inquiries can be directed to the corresponding author.

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
