# Peer review of "Improving the Enzymatic Activity and Stability of a Lytic Polysaccharide Monooxygenase"

_ijms, 2023, doi:10.3390/ijms24108963_

Round 1

Reviewer 1 Report

The manuscript entitled “Improving the Enzymatic Activity and Stability of a Lytic Polysaccharide Monooxygenase” deals with the optimization of protein sequence of Lytic Polysaccharide Monooxygenase from Bacillus amyloliquefaciens. The authors have shown that this procedure led to an increase in the enzyme activity for several substrates, in comparison to the original enzyme and in comparison, to cellulase. These results are important because this enzyme can be used in enzymatic mixtures to hydrolyze lignocellulosic materials for biofuel production, for example. There are important results reported, with the correct use of experimental procedures and conclusions are supported by data. It is well-written, and clear for a specialized audience. Minor corrections should be performed.

Abstract:

Line 15: species name in italics

Line 17: specify WT

Introduction

Line 82: O2, subscript 2

Last paragraph: In this paragraph it is supposed to say the goals, not the results or the conclusion of the work. Please attain to this.

Results

Section 2.1 some comments should be made in relation to the amino acid differences and the selection.

Figure 2: specify “Con” in figure legend. Specify abbreviations CMC, pNP and PNPC. Verify sentence (“against and…”) and punctuation of figure legend.

Line 160: specify PASC

Figure 3: Specify “Cont.”

Figure 4: verify legend (punctuation, words, etc.); T reesei in italics; specify “Cel”.

Reviewer 2 Report

The current manuscript is well written and deserves to be reported as of interest for a large scientific community. Small corrections should be performed in the text: space between values and units, names in italics, etc... One major point of correction is the determination of Km and kcat values for each mutant. It is crucial for having a full structure function relationships and to largely improve the quality of this manuscript. It will help having a deeper understanding and better conclusions.

Reviewer 3 Report

Dear Authors, 

Your manuscript is well, logically written and presents interesting results. 

I also determined LPMO activity using this method with DMP.  It is not an entirely ideal method, but readily available. 

I have no comments on the manuscript, I read it with interest and pleasure. 

On the other hand, it would be very good to include in the supplementary material a picture of the SDS -PAGE of the purified recombinant proteins. I too am curious about this image after electrophoresis . 

Reviewer 4 Report

Lytic polysaccharide monooxygenase from the bacterium Bacillus amyloliquefaciens was improved by protein engineering using the sequence consensus method. Single and double mutants were constructed, and a significant increase in LPMO activity was observed. The mutants and the WT enzyme were studied and compared under real process conditions. The subject is very interesting and up to date. The work is systematic and carefully planned. The only objection is that, in addition to thermal stability, the operational stability of mutants and WT was not compared. However, this is not mandatory.

I recommend the work for publication after a minor revision.

1.       The comparison of hydrolysis and oxidation activity of LPMO should be indicated by the numbers

2.       Line 175-176 - The statement that enzyme activity and concentration of reducing sugar  is a linear  is questionable. It is also incorrect to compare the product concentrations after 12 hours of reaction time with the initial reaction rates. Namely, the activity is measured during the period in which enzyme inactivation, inhibition by products and reverse reactions are neglected. After 12 hours, all these additional effects have an impact.

3.       Line 279-282 - According to this, it turns out that only the hydrolytic effect is responsible for the synergy of cellulolytic enzymes and LPMO. This is not correct, as most mutants also showed a significant increase in oxidation potential. Moreover, the oxidation potential of LPMO is significantly higher than the oxidation potential. It should be reformulated

4.       Line 302 - The recent work by Zhou et al. 2022. (doi: 10.3389/fbioe.2021.815990) considering the improvement of the stability of the enzyme should be cited.

5.       The conclusion is too long. The main results should be highlighted.

Round 2

Reviewer 2 Report

One major point of correction is the determination of Km and kcat values for each mutant. It is crucial for having a full structure function relationships and to largely improve the quality of this manuscript. It will help having a deeper understanding and better conclusions. Mutations from outside the active site is not a correct answer, as motion can change the recognition as kcat. Please perform the enzymology study before resubmitting the manuscript.

Round 3

Reviewer 2 Report

The authors have included the corrections. The manuscript desserves now to be reported.